# Learning Beam Search Policies via Imitation Learning

**Renato Negrinho**[1]      **Matthew R. Gormley**[1]      **Geoffrey J. Gordon**[1,2]
[1]Machine Learning Department, Carnegie Mellon University
[2]Microsoft Research
{negrinho,mgormley,ggordon}@cs.cmu.edu

## Abstract

Beam search is widely used for approximate decoding in structured prediction problems. Models often use a beam at test time but ignore its existence at train time, and therefore do not explicitly learn how to use the beam. We develop an unifying meta-algorithm for learning beam search policies using imitation learning. In our setting, the beam is part of the model, and not just an artifact of approximate decoding. Our meta-algorithm captures existing learning algorithms and suggests new ones. It also lets us show novel no-regret guarantees for learning beam search policies.

## 1 Introduction

Beam search is the dominant method for approximate decoding in structured prediction tasks such as machine translation [1], speech recognition [2], image captioning [3], and syntactic parsing [4]. Most models that use beam search at test time ignore the beam at train time and instead are learned via methods like likelihood maximization. They therefore suffer from two issues that we jointly address in this work: (1) learning ignores the existence of the beam and (2) learning uses only oracle trajectories. These issues lead to mismatches between the train and test settings that negatively affect performance. Our work addresses these two issues simultaneously by using imitation learning to develop novel beam-aware algorithms with no-regret guarantees. Our analysis is inspired by DAgger [5].

Beam-aware learning algorithms use beam search at both train and test time. These contrast with common two-stage learning algorithms that, first, at train time, learn a probabilistic model via maximum likelihood, and then, at test time, use beam search for approximate decoding. The insight behind beam-aware algorithms is that, if the model uses beam search at test time, then the model should be learned using beam search at train time. Resulting beam-aware methods run beam search at train time (i.e., roll-in) to collect losses that are then used to update the model parameters. The first proposed beam-aware algorithms are perceptron-based, updating the parameters either when the best hypothesis does not score first in the beam [6], or when it falls out of the beam [7].

While there is substantial prior work on beam-aware algorithms, none of the existing algorithms expose the learned model to its own consecutive mistakes at train time. When rolling in with the learned model, if a transition leads to a beam without the correct hypothesis, existing algorithms either stop [6, 8, 9] or reset to a beam with the correct hypothesis [7, 10, 11].[1] Additionally, existing beam-aware algorithms either do not have theoretical guarantees or only have perceptron-style guarantees [10]. We are the first to prove no-regret guarantees for an algorithm to learn beam search policies.

Imitation learning algorithms, such as DAgger [5], leverage the ability to query an oracle at train time to learn a model that is competitive (in the no-regret sense) to the best model in hindsight. Existing imitation learning algorithms such as SEARN [13], DAgger [5][2], AggreVaTe [15], and LOLS [16], execute the learned model at train time to collect data that is then labeled by the oracle and used for retraining. Nonetheless, these methods do not take the beam into account at train time, and therefore do not learn to use the beam effectively at test time.

We propose a new approach to learn beam search policies using imitation learning that addresses these two issues. We formulate the problem as learning a policy to traverse the combinatorial search space of beams. The learned policy is induced via a scoring function: the neighbors of the elements of a beam are scored and the top $k$ are used to form the successor beam. We learn a scoring function to match the ranking induced by the oracle costs of the neighbors. We introduce training losses that capture this insight, among which are variants of the weighted all pairs loss [17] and existing beam-aware losses. As the losses we propose are differentiable with respect to the scores, our scoring function can be learned using modern online optimization algorithms, e.g. Adam [18].

In some problems (e.g., sequence labeling and syntactic parsing) we have the ability to compute oracle completions and oracle completion costs for non-optimal partial outputs. Within our imitation learning framework, we can use this ability to compute oracle completion costs for the neighbors of the elements of a beam at train time to induce an oracle that allows us to continue collecting supervision after the best hypothesis falls out of the beam. Using this oracle information, we are able to propose a DAgger-like beam-aware algorithm with no-regret guarantees.

We describe our novel learning algorithm as an instantiation of a meta-algorithm for learning beam search policies. This meta-algorithm sheds light into key design decisions that lead to more performant algorithms, e.g., the introduction of better training losses. Our meta-algorithm captures much of the existing literature on beam-aware methods (e.g., [7, 8]), allowing a clearer understanding of and comparison to existing approaches, for example, by emphasizing that they arise from specific choices of training loss function and data collection strategy, and by proving novel regret guarantees for them.

Our contributions are: an algorithm for learning beam search policies (Section 4.2) with accompanying regret guarantees (Section 5), a meta-algorithm that captures much of the existing literature (Section 4), and new theoretical results for the early update [6] and LaSO [7] algorithms (Section 5.3).

## 2 Preliminaries

**Structured Prediction as Learning to Search**    We consider structured prediction in the learning to search framework [13, 5]. Input-output training pairs $D = \{(x_1, y_1), \ldots, (x_m, y_m)\}$ are drawn according to a data generating distribution $\mathcal{D}$ jointly over an input space $\mathcal{X}$ and an output space $\mathcal{Y}$. For each input $x \in \mathcal{X}$, there is an underlying search space $G_x = (V_x, E_x)$ encoded as a directed graph with nodes $V_x$ and edges $E_x$. Each output $y \in \mathcal{Y}_x$ is encoded as a terminal node in $G_x$, where $\mathcal{Y}_x \subseteq \mathcal{Y}$ is the set of valid structured outputs for $x$.

In this paper, we deal with stochastic policies $\pi : V_x \to \Delta(V_x)$, where $\Delta(V_x)$ is the set of probability distributions over nodes in $V_x$. (For convenience and brevity of presentation, we make our policies deterministic later in the paper through the introduction of a tie-breaking total order over the elements of $V_x$, but our arguments and theoretical results hold more generally.) The goal is to learn a stochastic policy $\pi(\cdot, x, \theta) : V_x \to \Delta(V_x)$ parametrized by $\theta \in \Theta \subseteq \mathbb{R}^p$ that traverses the induced search spaces, generating outputs with small expected cost; i.e., ideally, we would want to minimize

$$c(\theta) = \mathbb{E}_{(x,y)\sim\mathcal{D}}\mathbb{E}_{\hat{y}\sim\pi(\cdot,x,\theta)}c_{x,y}(\hat{y}), \tag{1}$$

where $c_{x,y} : \mathcal{Y}_x \to \mathbb{R}$ is the cost function comparing the ground-truth labeling $y$ to the predicted labeling $\hat{y}$. We are not able to optimize directly the loss in Equation (1), but we are able to find a mixture of policies $\theta_1, \ldots, \theta_m$, where $\theta_t \in \Theta$ for all $t \in [m]$, that is competitive with the best policy in $\Theta$ in the distribution of trajectories induced by the mixture of $\theta_1, \ldots, \theta_m$. We use notation $\hat{y} \sim \pi(\cdot, x, \theta)$ to mean that $\hat{y}$ is generated by sampling a trajectory $v_1, \ldots, v_h$ on $G_x$ by executing policy $\pi(\cdot, x, \theta)$, and returning the labeling $\hat{y} \in \mathcal{Y}$ associated with terminal node $v_h \in T$. The search spaces, cost functions and policies depend on $x \in \mathcal{X}$ or $(x, y) \in \mathcal{X} \times \mathcal{Y}$—in the sequel, we omit indexing by example for conciseness.

**Search Space, Cost, and Policies**  Each example $(x, y) \in \mathcal{X} \times \mathcal{Y}$ induces a search space $G = (V, E)$ and a cost function $c : \mathcal{Y} \to \mathbb{R}$. For all $v \in V$, we introduce its set of neighbors $N_v = \{v' \in V \mid (v, v') \in E\}$. We identify a single initial node $v_{(0)} \in V$. We define the set of terminal nodes $T = \{v \in V \mid N_v = \emptyset\}$. We assume without loss of generality that all nodes are reachable from $v_{(0)}$ and that all nodes have paths to terminal nodes. For clarity of exposition, we assume that $G$ is a tree-structured directed graph where all terminals nodes are at distance $h$ from the root $v_{(0)}$. We describe in Appendix A how to convert a directed graph search space to a tree-structured one with all terminals at the same depth.

Each terminal node $v \in T$ corresponds to a complete output $y \in \mathcal{Y}$, which can be compared to the ground-truth $y^* \in \mathcal{Y}$ via a cost function $c : T \to \mathbb{R}$ of interest (e.g., Hamming loss in sequence labeling or negative BLEU score [19] in machine translation). We define the optimal completion cost function $c^* : V \to \mathbb{R}$, which computes the cost of the best terminal node reachable from $v \in V$ as $c^*(v) = \min_{v' \in T_v} c(v')$, where $T_v$ is the set of terminal nodes reachable from $v$.

The definition of $c^* : V \to \mathbb{R}$ naturally gives rise to an oracle policy $\pi^*(\cdot, c^*) : V \to \Delta(V)$. At $v \in V$, $\pi^*(v, c^*)$ can be any fixed distribution (e.g., uniform or one-hot) over $\arg\min_{v' \in N_v} c^*(v')$. For any state $v \in V$, executing $\pi^*(\cdot, c^*)$ until arriving at a terminal node achieves the lowest possible cost for completions of $v$.

At $v \in V$, a greedy policy $\pi : V \to \Delta(V)$ induced by a scoring function $s : V \to \mathbb{R}$ computes a fixed distribution $\pi(v, \theta)$ over $\arg\max_{v' \in N_v} s(v', \theta)$. When multiple elements are tied with the same highest score, we can choose an arbitrary distribution over them. If there is a single highest scoring element, the policy is deterministic. In this paper, we assume the existence of a total order over the elements of $V$ that is used for breaking ties induced by a scoring function. The tie-breaking total ordering allows us to talk about a particular unique ordering, even when ties occur. The oracle policy $\pi^*(\cdot, c^*) : V \to \Delta(V)$ can be thought as being induced by the scoring function $-c^* : V \to \mathbb{R}$.

## 3  Beam search

**Beam Search Space**  Given a search space $G$, we construct its beam search space $G_k = (V_k, E_k)$, where $k \in \mathbb{N}$ is the maximum beam capacity. $V_k$ is the set of possible beams that can be formed along the search process, and $E_k$ is the set of possible beam transitions. Nodes $b \in V_k$ correspond to nonempty sets of nodes of $V$ with size upper bounded by $k$, i.e., $b = \{v_1, \ldots, v_{|b|}\}$ with $1 \leq |b| \leq k$ and $v_i \in V$ for all $i \in [|b|]$. The initial beam state $b_{(0)} \in V_k$ is the singleton set with the initial state $v_{(0)} \in V$. Terminal nodes in $T_k$ are singleton sets with a single terminal node $v \in T$. For $b \in V_k$, we define $A_b = \cup_{v \in b} N_v$, i.e., the union of the neighborhoods of the elements in $b$.

---

**Algorithm 1** Beam Search

1: **function** BEAMSEARCH$(G, k, \theta)$
2:    $b \leftarrow \{v_{(0)}\} \equiv b_{(0)}$
3:    **while** BEST$(b, 1, s(\cdot, \theta)) \notin T$ **do**
4:       $b \leftarrow$ POLICY$(G, b, k, s(\cdot, \theta))$
5:    **return** BEST$(b, 1, s(\cdot, \theta))$

6: **function** POLICY$(G, b, k, f)$
7:    Let $A_b = \cup_{v \in b} N_v$
8:    **return** BEST$(A_b, k, f)$

9: **function** BEST$(A, k, f)$
10:    Let $A = \{v_1, \ldots, v_n\}$ be ordered
11:       such that $f(v_1) \geq \cdots \geq f(v_n)$
12:    Let $k' = \min(k, n)$
13:    **return** $v_1, \ldots, v_{k'}$

---

Algorithm 1 describes the beam search variant used in our paper. In this paper, all elements in the beam are simultaneously expanded when transitioning. It is possible to define different beam search space variants, e.g., by considering different expansion strategies or by handling terminals differently (in the case where terminals can be at different depths) [3]. The arguments developed in this paper can be extended to those variants in a straightforward manner.

**Beam Costs**  We define the cost of a beam to be the cost of its lowest cost element, i.e., we have $c^* : V_k \to \mathbb{R}$ and, for $b \in V_k$, $c^*(b) = \min_{v \in b} c^*(v)$. We define the beam transition cost function $c : E_k \to \mathbb{R}$ to be $c(b, b') = c^*(b') - c^*(b)$, for $(b, b') \in E_k$, i.e., the difference in cost between the lowest cost element in $b'$ and the lowest cost element in $b$.

A cost increase occurs on a transition $(b, b') \in E_k$ if $c^*(b') > c^*(b)$, or equivalently, $c(b, b') > 0$, i.e., $b'$ dropped all the lowest cost neighbors of the elements of $b$. For all $b \in V_k$, we define

$N_b^* = \{b' \in N_b \mid c(b, b') = 0\}$, i.e., the set of beams neighboring $b$ that do not lead to cost increases. We will significantly overload notation, but usage will be clear from context and argument types, e.g., when referring to $c^* : V \to \mathbb{R}$ and $c^* : V_k \to \mathbb{R}$.

**Beam Policies** Let $\pi : V_k \to \Delta(V_k)$ be a policy induced by a scoring function $f : V \to \mathbb{R}$. To sample $b' \sim \pi(b)$ for a beam $b \in V_k$, form $A_b$, and compute scores $f(v)$ for all $v \in A_b$; let $v_1, \ldots, v_n$ be the elements of $A_b$ ordered such that $f(v_1) \geq \ldots \geq f(v_n)$; if $v_1 \in T$, $b' = \{v_1\}$; if $v_1 \notin T$, let $b'$ pick the $k$ top-most elements from $A_b \setminus T$. At $b \in V_k$, if there are many orderings that sort the scores of the elements of $A_b$, we can choose a single one deterministically or sample one stochastically; if there is a single such ordering, the policy $\pi : V_k \to \Delta(V_k)$ is deterministic at $b$.

For each $x \in \mathcal{X}$, at train time, we have access to the optimal path cost function $c^* : V \to \mathbb{R}$, which induces the oracle policy $\pi^*(\cdot, c^*) : V_k \to \Delta(V_k)$. At a beam $b$, a successor beam $b' \in N_b$ is optimal if $c^*(b') = c^*(b)$, i.e., at least one neighbor with the smallest possible cost was included in $b'$. The oracle policy $\pi^*(\cdot, c^*) : V_k \to \Delta(V_k)$ can be seen as using scoring function $-c^* : V_k \to \mathbb{R}$ to transition in the beam search space $G_k$.

---

**Algorithm 2** Meta-algorithm

1: **function** LEARN($D, \theta_1, k$)
2:      **for** each $t \in [|D|]$ **do**
3:          Induce $G$ using $x_t$
4:          Induce $s(\cdot, \theta_t) : V \to \mathbb{R}$ using $G$ and $\theta_t$
5:          Induce $c^* : V \to \mathbb{R}$ using $(x_t, y_t)$
6:          $b_{1:j} \leftarrow$ BEAMTRAJECTORY($G, c^*, s(\cdot, \theta_t), k$)
7:          Incur losses $\ell(\cdot, b_1), \ldots, \ell(\cdot, b_{j-1})$
8:          Compute $\theta_{t+1}$ using $\sum_{i=1}^{j-1} \ell(\cdot, b_i)$, e.g., by SGD or Adam
9:      **return** best $\theta_t$ on validation

10: **function** BEAMTRAJECTORY($G, c^*, f, k$)
11:      $b_1 \leftarrow \{v_{(0)}\} \equiv b_{(0)}$
12:      $j = 1$
13:      **while** BEST($b_j, 1, f) \notin T$ **do**
14:          **if** strategy is oracle **then**
15:              $b_{j+1} \leftarrow$ POLICY($G, b_j, k, -c^*$)
16:          **else**
17:              $b_{j+1} \leftarrow$ POLICY($G, b_j, k, f$)
18:              **if** $c^*(b_{j+1}) > c^*(b_j)$ **then**
19:                  **if** strategy is stop **then**
20:                      break
21:                  **if** strategy is reset **then**
22:                      $b_{j+1} \leftarrow$ POLICY($G, b_j, 1, -c^*$)
23:          $j \leftarrow j + 1$
24:      **return** $b_{1:j}$

---

## 4 Meta-Algorithm

Our goal is to learn a policy $\pi(\cdot, \theta) : V_k \to \Delta(V_k)$ induced by a scoring function $s(\cdot, \theta) : V \to \mathbb{R}$ that achieves small expected cumulative transition cost along the induced trajectories. Algorithm 2 presents our meta-algorithm in detail. Instantiating our meta-algorithm requires choosing both a surrogate training loss function (Section 4.1) and a data collection strategy (Section 4.2). Table 1 shows how existing algorithms can be obtained as instances of our meta-algorithm with specific choices of loss function, data collection strategy, and beam size.

### 4.1 Surrogate Losses

**Insight** In the beam search space, a prediction $\hat{y} \in \mathcal{Y}_x$ for $x \in \mathcal{X}$ is generated by running $\pi(\cdot, \theta)$ on $G_k$. This yields a beam trajectory $b_{1:h}$, where $b_1 = b_{(0)}$ and $b_h \in T_k$. We have

$$c(\theta) = \mathbb{E}_{(x,y) \sim \mathcal{D}} \mathbb{E}_{\hat{y} \sim \pi(\cdot, \theta)} c(\hat{y}) = \mathbb{E}_{(x,y) \sim \mathcal{D}} \mathbb{E}_{b_{1:h} \sim \pi(\cdot, \theta)} c^*(b_h). \tag{2}$$

The term $c^*(b_h)$ can be written in a telescoping manner as

$$c^*(b_h) = c^*(b_1) + \sum_{i=1}^{h-1} c(b_i, b_{i+1}). \tag{3}$$

As $c^*(b_1)$ depends on an example $(x, y) \in \mathcal{X} \times \mathcal{Y}$, but not on the parameters $\theta \in \Theta$, the set of minimizers of $c : \Theta \to \mathbb{R}$ is the same as the set of minimizers of

$$c'(\theta) = \mathbb{E}_{(x,y) \sim \mathcal{D}} \mathbb{E}_{b_{1:h} \sim \pi(\cdot, \theta)} \left( \sum_{i=1}^{h-1} c(b_i, b_{i+1}) \right). \tag{4}$$

It is not easy to minimize the cost function in Equation (4) as, for example, $c(b, \cdot) : V_k \to \mathbb{R}$ is combinatorial. To address this issue, we observe the following by using linearity of expectation and the law of iterated expectations to decouple the term in the sum over the trajectory:

$$\mathbb{E}_{b_{1:h} \sim \pi(\cdot, \theta)} \left( \sum_{i=1}^{h-1} c(b_i, b_{i+1}) \right) = \sum_{i=1}^{h-1} \mathbb{E}_{b_i \sim d_{\theta,i}} \mathbb{E}_{b_{i+1} \sim \pi(b_i, \theta)} c(b_i, b_{i+1})$$

$$= \mathbb{E}_{b_{1:h} \sim \pi(\cdot, \theta)} \left( \sum_{i=1}^{h-1} \mathbb{E}_{b' \sim \pi(b_i, \theta)} c(b_i, b') \right), \quad (5)$$

where $d_{\theta,i}$ denotes the distribution over beams in $V_k$ that results from following $\pi(\cdot, \theta)$ on $G_k$ for $i$ steps. We now replace $\mathbb{E}_{b' \sim \pi(b, \cdot)} c(b, b') : \Theta \to \mathbb{R}$ by a surrogate loss function $\ell(\cdot, b) : \Theta \to \mathbb{R}$ that is differentiable with respect to the parameters $\theta \in \Theta$, and where $\ell(\theta, b)$ is a surrogate loss for the expected cost increase incurred by following policy $\pi(\cdot, \theta)$ at beam $b$ for one step.

Elements in $A_b$ should be scored in a way that allows the best elements to be kept in the beam. Different surrogate losses arise from which elements we concern ourselves with, e.g., all the top $k$ elements in $A_b$ or simply one of the best elements in $A_b$. Surrogate losses are then large when the scores lead to discarding desired elements in $A_b$, and small when the scores lead to comfortably keeping the desired elements in $A_b$.

**Surrogate Loss Functions**   The following additional notation allows us to define losses precisely. Let $A_b = \{v_1, \ldots, v_n\}$ be an arbitrary ordering of the neighbors of the elements in $b$. Let $c = c_1, \ldots, c_n$ be the corresponding costs, where $c_i = c^*(v_i)$ for all $i \in [n]$, and $s = s_1, \ldots, s_n$ be the corresponding scores, where $s_i = s(v_i, \theta)$ for all $i \in [n]$. Let $\sigma^* : [n] \to [n]$ be a permutation such that $c_{\sigma^*(1)} \leq \ldots \leq c_{\sigma^*(n)}$, i.e., $v_{\sigma^*(1)}, \ldots, v_{\sigma^*(n)}$ are ordered in increasing order of cost. Note that $c^*(b) = c_{\sigma^*(1)}$. Similarly, let $\hat{\sigma} : [n] \to [n]$ be a permutation such that $s_{\hat{\sigma}(1)} \geq \ldots \geq s_{\hat{\sigma}(n)}$, i.e., $v_{\hat{\sigma}(1)}, \ldots, v_{\hat{\sigma}(n)}$ are ordered in decreasing order of score. We assume unique $\sigma^* : [n] \to [n]$ and $\hat{\sigma} : [n] \to [n]$ for simplifying the presentation of the loss functions (which can be guaranteed via the tie-breaking total order on $V$). In this case, at $b \in V_k$, the successor beam $b' \in N_b$ is uniquely determined by the scores of the elements of $A_b$.

For each $(x, y) \in \mathcal{X} \times \mathcal{Y}$, the corresponding cost function $c^* : V \to \mathbb{R}$ is independent of the parameters $\theta \in \Theta$. We define a loss function $\ell(\cdot, b) : \Theta \to \mathbb{R}$ at a beam $b \in V_k$ in terms of the oracle costs of the elements of $A_b$. We now introduce some well-motivated surrogate loss functions. Perceptron and large-margin inspired losses have been used in early update [6], LaSO [7], and BSO [11]. We also introduce two log losses.

**perceptron (first)**   Penalizes the lowest cost element in $A_b$ not being put at the top of the beam. When applied on the first cost increase, this is equivalent to an "early update" [6].

$$\ell(s, c) = \max \left( 0, s_{\hat{\sigma}(1)} - s_{\sigma^*(1)} \right). \quad (6)$$

**perceptron (last)**   Penalizes the lowest cost element in $A_b$ falling out of the beam.

$$\ell(s, c) = \max \left( 0, s_{\hat{\sigma}(k)} - s_{\sigma^*(1)} \right). \quad (7)$$

**margin (last)**   Prefers the lowest cost element to be scored higher than the last element in the beam by a margin. This yields updates that are similar but not identical to the approximate large-margin variant of LaSO [7].

$$\ell(s, c) = \max \left( 0, 1 + s_{\hat{\sigma}(k)} - s_{\sigma^*(1)} \right) \quad (8)$$

**cost-sensitive margin (last)**   Weights the margin loss by the cost difference between the lowest cost element and the last element in the beam. When applied on a LaSO-style cost increase, this is equivalent to the BSO update of [11].

$$\ell(s, c) = (c_{\hat{\sigma}(k)} - c_{\sigma^*(1)}) \max \left( 0, 1 + s_{\hat{\sigma}(k)} - s_{\sigma^*(1)} \right). \quad (9)$$

**upper bound** Convex upper bound to the expected beam transition cost, $\mathbb{E}_{b'\sim\pi(b,\cdot)}c(b,b') : \Theta \to \mathbb{R}$, where $b'$ is induced by the scores $s \in \mathbb{R}^n$.

$$\ell(s,c) = \max\left(0, \delta_{k+1}, \ldots, \delta_n\right) \tag{10}$$

where $\delta_j = (c_{\sigma^*(j)} - c_{\sigma^*(1)})(s_{\sigma^*(j)} - s_{\sigma^*(1)} + 1)$ for $j \in \{k+1, \ldots, n\}$. Intuitively, this loss imposes a cost-weighted margin between the best neighbor $v_{\sigma^*(1)} \in A_b$ and the neighbors $v_{\sigma^*(k+1)}, \ldots, v_{\sigma^*(n)} \in A_b$ that ought not to be included in the best successor beam $b'$. We prove in Appendix B that this loss is a convex upper bound for the expected beam transition cost.

**log loss (beam)** Normalizes only over the top $k$ neighbors of a beam according to the scores $s$.

$$\ell(s,c) = -s_{\sigma^*(1)} + \log\left(\sum_{i \in I} \exp(s_i)\right), \tag{11}$$

where $I = \{\sigma^*(1), \hat{\sigma}(1), \ldots, \hat{\sigma}(k)\}$. The normalization is only over the correct element $v_{\sigma^*(1)}$ and the elements included in the beam. The set of indices $I \subseteq [n]$ encodes the fact that the score vector $s \in \mathbb{R}^n$ may not place $v_{\sigma^*(1)}$ in the top $k$, and therefore it has to also be included in that case. This loss is used in [9], albeit introduced differently.

**log loss (neighbors)** Normalizes over all elements in $A_b$.

$$\ell(s,c) = -s_{\sigma^*(1)} + \log\left(\sum_{i=1}^{n} \exp(s_i)\right) \tag{12}$$

**Discussion** The losses here presented directly capture the purpose of using a beam for prediction— ensuring that the best hypothesis stays in the beam, i.e., that, at $b \in V_k$, $v_{\sigma^*(1)} \in A_b$ is scored sufficiently high to be included in the successor beam $b' \in N_b$. If full cost information is not accessible, i.e., if are not able to evaluate $c^* : V \to \mathbb{R}$ for arbitrary elements in $V$, it is still possible to use a subset of these losses, provided that we are able to identify the lowest cost element among the neighbors of a beam, i.e., for all $b \in V_k$, an element $v \in A_b$, such that $c^*(v) = c^*(b)$.

While certain losses do not appear beam-aware (e.g., those in Equation (6) and Equation (12)), it is important to keep in mind that all losses are collected by executing a policy on the beam search space $G_k$. Given a beam $b \in V_k$, the score vector $s \in \mathbb{R}^n$ and cost vector $c \in \mathbb{R}^n$ are defined for the elements of $A_b$. The losses incurred depend on the specific beams visited. Losses in Equation (6), (10), and (12) are convex. The remaining losses are non-convex. For $k = 1$, we recover well-known losses, e.g., loss in Equation (12) becomes a simple log loss over the neighbors of a single node, which is precisely the loss used in typical log-likelihood maximization models; loss in Equation (7) becomes a perceptron loss. In Appendix C we discuss convexity considerations for different types of losses. In Appendix D, we present additional losses and expand on their connections to existing work.

## 4.2 Data Collection Strategy

Our meta-algorithm requires choosing a train time policy $\pi : V_k \to \Delta(V_k)$ to traverse the beam search space $G_k$ to collect supervision. Sampling a trajectory to collect training supervision is done by BEAMTRAJECTORY in Algorithm 2.

**oracle** Our simplest policy follows the oracle policy $\pi^* : V_k \to \Delta(V_k)$ induced by the optimal completion cost function $c^* : V \to \mathbb{R}$ (as in Section 3). Using the terminology of Algorithm 1, we can write $\pi^*(b, c^*) = \text{POLICY}(G, b, k, -c^*)$. This policy transitions using the negated sorted costs of the elements in $A_b$ as scores.

The oracle policy does not address the distribution mismatch problem. At test time, the learned policy will make mistakes and visit beams for which it has not collected supervision at train time, leading to error compounding. Imitation learning tells us that it is necessary to collect supervision at train time with the learned policy to avoid error compounding at test time [5].

We now present data collection strategies that use the learned policy. For brevity, we only cover the case where the learned policy is always used (except when the transition leads to a cost-increase), and leave the discussion of additional possibilities (e.g., probabilistic interpolation of learned and oracle policies) to Appendix E.3. When an edge $(b, b') \in E_k$ incurring cost increase is traversed, different strategies are possible:

Table 1: Existing and novel beam-aware algorithms as instances of our meta-algorithm. Our theoretical guarantees require the existence of a deterministic no-regret online learning algorithm for the resulting problem.

| Algorithm | Meta-algorithm choices | | |
|---|---|---|---|
| | data collection | surrogate loss | $k$ |
| log-likelihood | oracle | log loss (neighbors) | 1 |
| DAGGER [5] | continue | log loss (neighbors) | 1 |
| early update [6] | stop | perceptron (first) | $> 1$ |
| LaSO (perceptron) [7] | reset | perceptron (first) | $> 1$ |
| LaSO (large-margin) [7] | reset | margin (last) | $> 1$ |
| BSO [11] | reset | cost-sensitive margin (last) | $> 1$ |
| globally normalized [9] | stop | log loss (beam) | $> 1$ |
| Ours | continue | [choose a surrogate loss] | $> 1$ |

**stop** Stop collecting the beam trajectory. The last beam in the trajectory is $b'$, i.e., the beam on which we arrive in the transition that led to a cost increase. This data collection strategy is used in structured perceptron training with early update [6].

**reset** Reset the beam to contain only the best state as defined by the optimal completion cost function: $b' = \text{BEST}(b, 1, -c^*)$. In the subsequent steps of the policy, the beam grows back to size $k$. LaSO [7] uses this data collection strategy. Similarly to the oracle data collection strategy, rather than committing to a specific $b' \in N_b^*$, we can sample $b' \sim \pi^*(b, c^*)$ where $\pi^*(b, c^*)$ is any distribution over $N_b^*$. The reset data collection strategy collects beam trajectories where the oracle policy $\pi$ is executed conditionally, i.e., when the roll-in policy $\pi(\cdot, \theta_t)$ would lead to a cost increase.

**continue** We can ignore the cost increase and continue following policy $\pi_t$. This is the strategy taken by DAgger [5]. The continue data collection strategy has not been considered in the beam-aware setting, and therefore it is a novel contribution of our work. Our stronger theoretical guarantees apply to this case.

## 5 Theoretical Guarantees

We state regret guarantees for learning beam search policies using the continue, reset, or stop data collection strategies. One of the main contributions of our work is framing the problem of learning beam search policies in a way that allows us to obtain meaningful regret guarantees. Detailed proofs are provided in Appendix E. We begin by analyzing the continue collection strategy. As we will see, regret guarantees are stronger for continue than for stop or reset.

No-regret online learning algorithms have an important role in the proofs of our guarantees. Let $\ell_1, \ldots, \ell_m$ be a sequence of loss functions with $\ell_t : \Theta \to \mathbb{R}$ for all $t \in [m]$. Let $\theta_1, \ldots, \theta_m$ be a sequence of iterates with $\theta_t \in \Theta$ for all $t \in [m]$. The loss function $\ell_t$ can be chosen according to an arbitrary rule (e.g., adversarially). The online learning algorithm chooses the iterate $\theta_t$. Both $\ell_t$ and $\theta_t$ are chosen online, as functions of loss functions $\ell_1, \ldots, \ell_{t-1}$ and iterates $\theta_1, \ldots, \theta_{t-1}$.

**Definition 1.** *An online learning algorithm is no-regret if for any sequence of functions $\ell_1, \ldots, \ell_m$ chosen according to the conditions above we have*

$$\frac{1}{m} \sum_{t=1}^{m} \ell_t(\theta_t) - \min_{\theta \in \Theta} \frac{1}{m} \sum_{t=1}^{m} \ell_t(\theta) = \gamma_m, \tag{13}$$

*where $\gamma_m$ goes to zero as $m$ goes to infinity.*

Many no-regret online learning algorithms, especially for convex loss functions, have been proposed in the literature, e.g., [20, 21, 22]. Our proofs of the theoretical guarantees require the no-regret online learning algorithm to be deterministic, i.e., $\theta_t$ to be a deterministic rule of previous observed iterates $\theta_1, \ldots, \theta_{t-1}$ and loss functions $\ell_1, \ldots, \ell_{t-1}$, for all $t \in [m]$. Online gradient descent [20] is an example of such an algorithm.

In Theorem 1, we prove no-regret guarantees for the case where the no-regret online algorithm is presented with explicit expectations for the loss incurred by a beam search policy. In Theorem 2, we upper bound the expected cost incurred by a beam search policy as a function of its expected loss. This result holds in cases where, at each beam, the surrogate loss is an upper bound on the expected cost increase at that beam. In Theorem 3, we use Azuma-Hoeffding to prove no-regret high probability bounds for the case where we only have access to empirical expectations of the loss incurred by a policy, rather than explicit expectations. In Theorem 4, we extend Theorem 3 for the case where the data collection policy is different from the policy that we are evaluating. These results allow us to give regret guarantees that depend on how frequently is the data collection policy different from the policy that we are evaluating.

In this section we simply state the results of the theorems alongside some discussion. All proofs are presented in detail in Appendix E. Our analysis closely follows that of DAgger [5], although the results need to be interpreted in the beam search setting. Our regret guarantees for beam-aware algorithms with different data collection strategies are novel.

## 5.1 No-Regret Guarantees with Explicit Expectations

The sequence of functions $\ell_1, \ldots, \ell_m$ can be chosen in a way that applying a no-regret online learning algorithm to generate the sequence of policies $\theta_1, \ldots, \theta_m$ leads to no-regret guarantees for the performance of the mixture of $\theta_1, \ldots, \theta_m$. The adversary presents the no-regret online learning algorithm with $\ell_t = \ell(\cdot, \theta_t)$ at time $t \in [m]$. The adversary is able to play $\ell(\cdot, \theta_t)$ because it can anticipate $\theta_t$, as the adversary knows the deterministic rule used by the no-regret online learning algorithm to pick iterates. Paraphrasing Theorem 1, on the distribution of trajectories induced by the the uniform stochastic mixture of $\theta_1, \ldots, \theta_m$, the best policy in $\Theta$ for this distribution performs as well (in the limit) as the uniform mixture of $\theta_1, \ldots, \theta_m$.

**Theorem 1.** *Let* $\ell(\theta, \theta') = \mathbb{E}_{(x,y)\sim\mathcal{D}}\mathbb{E}_{b_{1:h}\sim\pi(\cdot,\theta')}\left(\sum_{i=1}^{h-1}\ell(\theta, b_i)\right)$. *If the sequence* $\theta_1, \ldots, \theta_m$ *is chosen by a deterministic no-regret online learning algorithm, we have* $\frac{1}{m}\sum_{t=1}^{m}\ell(\theta_t, \theta_t) - \min_{\theta\in\Theta}\frac{1}{m}\sum_{t=1}^{m}\ell(\theta, \theta_t) = \gamma_m$, *where* $\gamma_m$ *goes to zero when* $m$ *goes to infinity.*

Furthermore, if for all $(x, y) \in \mathcal{X} \times \mathcal{Y}$ the surrogate loss $\ell(\cdot, b) : \Theta \to \mathbb{R}$ is an upper bound on the expected cost increase $\mathbb{E}_{b'\sim\pi(b,\cdot)}c(b, b') : \Theta \to \mathbb{R}$ for all $b \in V_k$, we can transform the surrogate loss no-regret guarantees into performance guarantees in terms of $c : \mathcal{Y} \to \mathbb{R}$. Theorem 2 tells us that if the best policy along the trajectories induced by the mixture of $\theta_1, \ldots, \theta_m$ in $\Theta$ incurs small surrogate loss, then the expected cost resulting from labeling examples $(x, y) \in \mathcal{X} \times \mathcal{Y}$ sampled from $\mathcal{D}$ with the uniform mixture of $\theta_1, \ldots, \theta_m$ is also small. It is possible to transform the results about the uniform mixture of $\theta_1, \ldots, \theta_m$ on results about the best policy among $\theta_1, \ldots, \theta_m$, e.g., following the arguments of [23], but for brevity we do not present them in this paper. Proofs of Theorem 1 and Theorem 2 are in Appendix E.1

**Theorem 2.** *Let all the conditions in Definition 1 be satisfied. Additionally, let* $c(\theta) = c^*(b_1) + \mathbb{E}_{(x,y)\sim\mathcal{D}}\mathbb{E}_{b_{1:h}\sim\pi(\cdot,\theta)}\left(\sum_{i=1}^{h-1}c(b_i, b_{i+1})\right) = \mathbb{E}_{(x,y)\sim\mathcal{D}}\mathbb{E}_{b_{1:h}\sim\pi(\cdot,\theta)}c^*(b_h)$. *Let* $\ell(\cdot, b) : \Theta \to \mathbb{R}$ *be an upper bound on* $\mathbb{E}_{b'\sim\pi(b,\cdot)}c(b, b') : \Theta \to \mathbb{R}$, *for all* $b \in V_k$. *Then,* $\frac{1}{m}\sum_{t=1}^{m}c(\theta_t) \leq \mathbb{E}_{(x,y)\sim\mathcal{D}}c^*(b_1) + \min_{\theta\in\Theta}\frac{1}{m}\sum_{t=1}^{m}\ell(\theta, \theta_t) + \gamma_m$, *where* $\gamma_m$ *goes to zero as* $m$ *goes to infinity.*

## 5.2 Finite Sample Analysis

Theorem 1 and Theorem 2 are for the case where the adversary presents explicit expectations, i.e., the loss function at time $t \in [m]$ is $\ell_t(\cdot) = \mathbb{E}_{(x,y)\sim\mathcal{D}}\mathbb{E}_{b_{1:h}\sim\pi(\cdot,\theta_t)}\left(\sum_{i=1}^{h-1}\ell(\cdot, b_i)\right)$. We most likely only have access to a sample estimator $\hat{\ell}(\cdot, \theta_t) : \Theta \to \mathbb{R}$ of the true expectation: we first sample an example $(x_t, y_t) \sim \mathcal{D}$, sample a trajectory $b_{1:h}$ according to $\pi(\cdot, \theta_t)$, and obtain $\hat{\ell}(\cdot, \theta_t) = \sum_{i=1}^{h-1}\ell(\cdot, b_i)$. We prove high probability no-regret guarantees for this case. Theorem 3 tells us that the population surrogate loss of the mixture of policies $\theta_1, \ldots, \theta_m$ is, with high probability, not much larger than its empirical surrogate loss. Combining this result with Theorem 1 and Theorem 2 allows us to give finite sample high probability results for the performance of the mixture of policies $\theta_1, \ldots, \theta_m$. The proof of Theorem 3 is found in Appendix E.2.

**Theorem 3.** *Let $\hat{\ell}(\cdot, \theta') = \sum_{i=1}^{h-1} \ell(\cdot, b_i)$ which is generated by sampling $(x, y)$ from $\mathcal{D}$ (which induces the corresponding beam search space $G_k$ and cost functions), and sampling a beam trajectory using $\pi(\cdot, \theta')$. Let $|\sum_{i=1}^{h-1} \ell(\theta, b_i)| \leq u$ for a constant $u \in \mathbb{R}$, for all $(x, y) \in \mathcal{X} \times \mathcal{Y}$, beam trajectories $b_{1:h}$, and $\theta \in \Theta$. Let the iterates be chosen by a no-regret online learning algorithm, based on the sequence of losses $\ell_t = \hat{\ell}(\cdot, \theta_t) : \Theta \to \mathbb{R}$, for $t \in [m]$, then we have $\mathbb{P}\left(\frac{1}{m}\sum_{t=1}^{m} \ell(\theta_t, \theta_t) \leq \frac{1}{m}\sum_{t=1}^{m} \hat{\ell}(\theta_t, \theta_t) + \eta(\delta, m)\right) \geq 1 - \delta$, where $\delta \in (0, 1]$ and $\eta(\delta, m) = u\sqrt{2\log(1/\delta)/m}$.*

### 5.3 Finite Sample Analysis for Arbitrary Data Collection Policies

All the results stated so far are for the continue data collection strategy where, at time $t \in [m]$, the whole trajectory $b_{1:h}$ is collected using the current policy $\pi(\cdot, \theta_t)$. Stop and reset data collection strategies do not necessarily collect the full trajectory under $\pi(\cdot, \theta_t)$. If the data collection policy $\pi' : V_k \to \Delta(V_k)$ is other than the learned policy, the analysis can be adapted by accounting for the difference in distribution of trajectories induced by the learned policy and the data collection policy. The insight is that $\sum_{i=1}^{h-1} \ell(\theta, b_i)$ only depends on $b_{1:h-1}$, so if no cost increases occur in this portion of the trajectory, we are effectively sampling the trajectory using $\pi(\cdot, \theta)$ when using the stop and reset data collection strategies.

Prior work presented only perceptron-style results for these settings [6, 7]—we are the first to present regret guarantees. Our guarantee depends on the probability with which $b_{1:h-1}$ is collected solely with $\pi(\cdot, \theta)$. We state the finite sample analysis result for the case where these probabilities are not known explicitly, but we are able to estimate them. The proof of Theorem 4 is found in Appendix E.3.

**Theorem 4.** *Let $\pi_t : V_k \to \Delta(V_k)$ be the data collection policy for example $t \in [m]$, which uses either the stop or reset data collection strategies. Let $\hat{\alpha}(\theta_t)$ be the empirical estimate of the probability of $\pi(\cdot, \theta_t)$ incurring at least one cost increase up to time $h - 1$. Then,*

$$\mathbb{P}\left(\frac{1}{m}\sum_{t=1}^{m} \ell(\theta_t, \theta_t) \leq \frac{1}{m}\sum_{t=1}^{m} \hat{\ell}(\theta_t, \pi_t) + u\left(1 - \frac{1}{m}\sum_{t=1}^{m} \hat{\alpha}(\theta_t)\right) + 2\eta(\delta, m)\right) \geq 1 - \delta,$$

*where $\delta \in (0, 1]$ and $\eta(\delta, m) = u\sqrt{2\log(1/\delta)/m}$.*

If the probability of stopping or resetting goes to zero as $m$ goes to infinity, then the term captures the discrepancy between the distributions of induced by $\pi(\cdot, \theta_t)$ and $\pi_t$ vanishes, and we recover a guarantee similar to Theorem 3. If the probability of stopping or resetting does not go completely to zero, it is still possible to provide regret guarantees for the performance of this algorithm but now with a term that does not vanish with increasing $m$. These regret guarantees for the different data collection strategies are novel.

## 6 Conclusion

We propose a framework for learning beam search policies using imitation learning. We provide regret guarantees for both new and existing algorithms for learning beam search policies. One of the main contributions is formulating learning beam search policies in the learning to search framework. Policies for beam search are induced via a scoring function. The intuition is that the best neighbors in a beam should be scored sufficiently high, allowing them to be kept in the beam when transitioning using these scores. Based on this insight, we motivate different surrogate loss functions for learning scoring functions. We recover existing algorithms in the literature through specific choices for the loss function and data collection strategy. Our work is the first to provide a beam-aware algorithm with no-regret guarantees.

#### Acknowledgments

The authors would like to thank Ruslan Salakhutdinov, Akshay Krishnamurthy, Wen Sun, Christoph Dann, and Kin Olivares for helpful discussions and detailed reviews.

## Footnotes

[1][12] take a different approach by training with a differentiable approximation of beam search, but decode with the standard (non-differentiable) search algorithm at test time.

[2] Scheduled sampling [14] is an instantiation of DAgger.

[3][13] mention the possibility of encoding complex search algorithms by defining derived search spaces.

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
