[Supplementary Material]

# A   Conversion to Tree-Structured Search Spaces

We define a search space as an arbitrary finite directed graph $G = (V, E)$, where $V$ is the set of nodes and $E \subset V \times V$ is the set of directed edges. Every directed graph $G = (V, E)$ has associated a tree-structured directed graph $G_p = (V_p, E_p)$ encoding all possible paths through $G$. An important reason to do this transformation is that, in practice, policies often incorporate history features, so they are functions of the whole path leading to a node in $G$, rather than just a single node in $G$. A policy becomes a function of single nodes of $G_p$. If $G$ is tree-structured, $G_p$ is isomorphic to $G$, i.e., they are the same search space.

The set of terminal nodes $T_p$ contains all paths from the initial node $v_{(0)} \in V$ to terminal nodes $v \in T$. For $v \in V_p$, we denote the length of the sequence encoding a path by $|v|$. The length of a path $v \in V_p$ is $|v| - 1$. We write $v_i$ for the $i$-th element of a path $v \in V_p$. For all $v \in V$, $v_i \in V$ for all $i \in [|v|]$ and $v_1 = v_{(0)}$. The sets $N_{p,v}, R_{p,v}, T_{p,v}$ for $v \in V_p$ are defined analogously to the sets $N_v, R_v, T_v$ for $v \in V$. For a path $v \in V_p$, $v' \in N_{p,v}$ if $v'_{1:|v|} = v$, $|v| = |v'| - 1$, and $v'_{|v'|} \in N_{v_{|v|}}$, i.e., a path $v' \in V_p$ neighbors $v \in V_p$ if it can be written as $v$ followed by an additional node in $N_{v_{|v|}}$. For $v \in V_p$, $v' \in R_{p,v}$ if $v$ is a prefix of $v'$ and $v' \in T_{p,v}$ if $v$ is a prefix of $v'$ and $v'_{|v'|} \in T$. As $G_p$ is tree-structured, we can define the depth $d_v$ of a path $v \in V_p$ as its length, i.e., $d_v = |v| - 1$. If path $v \in V_p$, then prefix $v_{1:i} \in V_p$, for all $i \in [|v|]$, i.e., path prefixes are themselves paths.

Tree-structured search spaces are common in practice. They often occur in write-only search spaces, where once an action is taken, its effects are irreversible. Typical search spaces for sequence tagging and machine translation are tree-structured: given a sequence to tag or translate, at each step we commit to a token and never get to change it. When the search space $G$ is not naturally seen as being tree-structured, the construction described makes it natural to work with an equivalent tree-structured search space of paths $G_p$.

If $G$ has cycles, $G_p$ would be infinite. Infinite cycling in $G_p$ can be prevented by, for example, introducing a maximum path length or a maximum number of times that any given node $v \in V$ can be visited. In this paper, we also assumed that all nodes in $T_p$ have distance $h$ to the root. It is possible to transform $G_p$ into a new tree-structured graph $G'_p$ by padding shorter paths to length $h$. Let $h$ be the maximum distance of any terminal in $T_p$ to the root. For each terminal node $v \in T_p$ with distance $d_v < h$ to the root, we extend the path to $v$ by appending a linear chain of $h - d_v$ additional nodes. Node $v$ is no longer a terminal node in $G'_p$, and all the nodes in $G'_p$ that resulted from extending the path are identified with $v$.

# B   Convex Upper Bound Surrogate for Expected Beam Transition Cost

In this appendix, we design a convex upper bound surrogate loss $\ell(\cdot, b) : \Theta \to \mathbb{R}$ for the expected beam transition cost $\mathbb{E}_{b' \sim \pi(b, \cdot)} c(b, b') : \Theta \to \mathbb{R}$. Let $A_b = \{v_1, \dots, v_n\}$ be an arbitrary ordering of the neighbors of $b$, with corresponding costs $c_1, \dots, c_n$, with $c_i = c^*(v_i)$ for all $i \in [n]$. Let $s_1, \dots, s_n$ be the corresponding scores, with $s_i = s(v_i, \theta)$ for all $i \in [n]$. Let $\sigma^* : [n] \to [n]$ and $\hat{\sigma} : [n] \to [n]$ be the unique permutations such that $c_{\sigma^*(1)} \le \dots \le c_{\sigma^*(n)}$ and $s_{\hat{\sigma}(1)} \ge \dots \ge s_{\hat{\sigma}(n)}$, respectively, with ties broken according to the total order on $V$. We have $c^*(b) = c_{\sigma^*(1)}$. Let $k \in \mathbb{N}$ be the maximum beam capacity. Let $b'$ be the beam induced by the scores $s_1, \dots, s_n$, i.e., $b' = \{v_{\hat{\sigma}(1)}, \dots, v_{\hat{\sigma}(k')}\}$, with $k' = \min(k, n)$ and ties broken according to the total order.

Consider the *upper bound* loss function (repeated here from Equation (10))

$$\ell(s, c) = \max\left(0, \delta_{k+1}, \dots, \delta_n\right), \tag{14}$$

where $\delta_j = (c_{\sigma^*(j)} - c_{\sigma^*(1)})(s_{\sigma^*(j)} - s_{\sigma^*(1)} + 1)$ for $j \in \{k+1, \dots, n\}$.

This loss function is lower bounded by zero, so we only need to show that it upper bounds $c(b, b')$ when there is a cost increase, i.e., when $c(b, b') > 0$. A cost increase $c(b, b') > 0$ implies that the best element $v_{\sigma^*(1)}$ fell off the beam, meaning that $b' = \{v_{\hat{\sigma}(1)}, \dots, v_{\hat{\sigma}(k)}\} \ne \{v_{\sigma^*(1)}, \dots, v_{\sigma^*(k)}\}$, and therefore $b' \cap \{v_{\sigma^*(k+1)}, \dots, v_{\sigma^*(n)}\} \ne \emptyset$. Let $v_{\sigma^*(j)} \in b' \cap \{v_{\sigma^*(k+1)}, \dots, v_{\sigma^*(n)}\}$, then

$s_{\sigma^*(j)} \geq s_{\sigma^*(1)}$ and $c(b, b') \leq c_{\sigma^*(j)} - c_{\sigma^*(1)}$, with $j \in \{k+1, \ldots, n\}$. We have

$$
\begin{aligned}
\max\left(0, \delta_{k+1}, \ldots, \delta_n\right) &\geq \delta_j \\
&= (c_{\sigma^*(j)} - c_{\sigma^*(1)})(s_{\sigma^*(j)} - s_{\sigma^*(1)} + 1) \\
&\geq c_{\sigma^*(j)} - c_{\sigma^*(1)} \\
&\geq c(b, b'),
\end{aligned}
$$

proving the upper bound property of the loss in Equation (14).

This loss is the maximum of a finite number of affine functions of the scores, and therefore convex with respect to the score vector $s \in \mathbb{R}^n$. The resulting optimization problem is convex with respect to the parameters of the scoring function if, for example, the scoring function is linear with respect to the parameters $\theta \in \Theta$, i.e., $s(v, \theta) = \theta^T \phi(v, x)$, where $\phi : V \times \mathcal{X} \to \mathbb{R}^p$ is a fixed feature function of the state. If $A_b$ has no more than $k$ elements, this surrogate loss is identically zero, i.e., for $k \geq n$, $\ell(s, c) = 0$, for all $s \in \mathbb{R}^n$ and $c \in \mathbb{R}^n$. If $k = 1$, we recover a greedy decoding algorithm and the loss in Equation (14) becomes a weighted hinge loss.

## C Convexity Considerations for Surrogate Loss Functions

It is common in the literature to update the parameters only when a cost increase occurs [10, 8, 9]. We show that the resulting loss surrogate functions are, in general, non-convex in the scores.

The following loss is an upper bound on the beam transition loss $c : E_k \to \mathbb{R}$, but is non-convex in the scores:

$$
\ell(s, c) = (c_{\hat{\sigma}(k)} - c_{\sigma^*(1)}) \max(0, s_{\hat{\sigma}(k)} - s_{\sigma^*(1)} + 1). \tag{15}
$$

The upper bound property for this loss is easy to verify: if $s \in \mathbb{R}^n$ at $b \in V_k$ induces $b' \in V_k$ with $c(b, b') > 0$, then $s_{\hat{\sigma}(k)} \geq s_{\sigma^*(1)}$ and $c_{\hat{\sigma}(k)} > c_{\sigma^*(1)}$, leading to

$$
\begin{aligned}
(c_{\hat{\sigma}(k)} - c_{\sigma^*(1)}) \max(0, s_{\hat{\sigma}(k)} - s_{\sigma^*(1)} + 1) &\geq c_{\hat{\sigma}(k)} - c_{\sigma^*(1)} \\
&\geq c(b, b'),
\end{aligned}
$$

as $v_{\hat{\sigma}(k)} \in b'$. This loss is used in [11]. The same reasoning holds when substituting $k$ in Equation (15) by any $i \in [k]$.

We now show that two aspects commonly present in the beam-aware literature lead to non-convexity of the surrogate losses. The first aspect is updating the parameters only when there is a cost increase. This amounts to defining a new loss function $\ell' : \mathbb{R}^n \times \mathbb{R}^n \to \mathbb{R}$ from $\ell : \mathbb{R}^n \times \mathbb{R}^n \to \mathbb{R}$ of the form

$$
\ell'(s, c) = \ell(s, c) \mathbb{1}[c(b, b') > 0],
$$

where $b'$ is induced by $s \in \mathbb{R}^n$. The second aspect that leads to non-convexity is indexing the score vector $s \in \mathbb{R}^n$ or cost vector $c \in \mathbb{R}^n$ with a function of the parameters, e.g., permutation $\hat{\sigma} : [n] \to [n]$ depends on the scores $s \in \mathbb{R}^n$ and therefore, on the parameters $\theta \in \Theta$. We show non-convexity with respect to the scores through two simple counter examples.

For the first aspect, let $k = 2$ and $n = 3$, with $v_1, v_2, v_3$ having costs $c_1 = 0, c_2 = 1, c_3 = 1$. Any beam that keeps $v_1$ has no cost increase. Consider the scores $s_1 = 1, s_2 = 10, s_3 = 0$ and $s_1' = 1, s_2' = 0, s_3' = 10$. Both $s$ and $s'$ lead to no cost increase, as both score vectors keep $v_1$ in the beam. For $\ell' : \mathbb{R}^n \times \mathbb{R}^n \to \mathbb{R}$ to be convex in the scores, we must have $\ell'(\alpha s + (1 - \alpha)s', c) \leq \alpha \ell'(s, c) + (1 - \alpha)\ell'(s', c)$, for all $\alpha \in [0, 1]$. As both $s$ and $s'$ lead to no cost increase, we have $\ell'(s, c) = \ell'(s', c) = 0$, yielding the following necessary condition for convexity: $\ell(\alpha s + (1 - \alpha)s', c) \leq 0$ for all $\alpha \in [0, 1]$. For $\alpha = 0.5$, we have $\bar{s}_1 = 1, \bar{s}_2 = 5, \bar{s}_3 = 5$, which leads to a cost increase, and therefore to loss $\ell'(\bar{s}, c) > 0$, implying that $\ell' : \mathbb{R}^n \times \mathbb{R}^n \to \mathbb{R}$ is non-convex in the scores.

For the second aspect, consider the loss in Equation (15). Ignore the multiplicative term involving the costs and consider only the hinge part $\max(0, s_{\hat{\sigma}(k)} - s_{\sigma^*(k)} + 1)$. Let $k = 2$ and $n = 3$. Consider that the elements $v_1, v_2, v_3$ are sorted in increasing order of cost; let $s_1 = 2, s_2 = 1, s_3 = 0$, and $s_1' = 2, s_2' = 4, s_3' = 0$. In both cases, the hinge part of loss in Equation (15) is zero, but if we take a convex combination of the scores with $\alpha = 0.5$, we get $\bar{s}_1 = 2, \bar{s}_2 = 2.5, \bar{s}_3 = 0$, for which the surrogate loss is nonzero (assuming that the costs of $v_1, v_2, v_3$ are unique).

# D  Additional Loss Functions

We present additional loss functions that were omitted in Section 4.1 and discuss their connections to previous work.

**cost sensitive margin (beam)**  Prefers the lowest cost element to be scored higher than best runner-up in the beam by a cost-weighted margin. With unbounded beam capacity, we recover the structured max-margin loss of [24] for $M^3Ns$.

$$\ell(s, c) = -s_{\sigma^*(1)} + \max_{i \in \{1,\ldots,k\}} \left( c_{\hat{\sigma}(i)} + s_{\hat{\sigma}(i)} \right) \tag{16}$$

**softmax margin (beam)**  Log loss that can be understood as smoothing the $\max$ in *cost sensitive margin (beam)*. With unbounded beam capacity, we recover the softmax-margin loss of [25] for CRFs.

$$\ell(s, c) = -s_{\sigma^*(1)} + \log \left( \sum_{i=1}^{k} \exp \left( c_{\hat{\sigma}(i)} + s_{\hat{\sigma}(i)} \right) \right) \tag{17}$$

**weighted pairs (all)**  Reduces the problem of producing the correct ranking over the neighbors to $n(n-1)/2$ weighted binary classification problems. Hinge terms for pairs with the same cost cancel, effectively expressing that we are indifferent to the relative order of the elements of the pair.

$$\ell(s, c) = \sum_{i=1}^{n} \sum_{j=i+1}^{n} \left( c_{\sigma^*(j)} - c_{\sigma^*(i)} \right) \max \left( 0, s_{\sigma^*(j)} - s_{\sigma^*(1)} + 1 \right) \tag{18}$$

**weighted pairs (bipartite)**  Only weighted pairs between elements than ought to be included in the beam and those that ought to excluded from the beam. A similar loss has been proposed for bipartite ranking, where the goal is to order all positive examples before all negative examples

$$\ell(s, c) = \sum_{i=1}^{k} \sum_{j=k+1}^{n} \left( c_{\sigma^*(j)} - c_{\sigma^*(i)} \right) \max \left( 0, s_{\sigma^*(j)} - s_{\sigma^*(1)} + 1 \right) \tag{19}$$

**weighted pairs (hybrid)**  Similar to weighted pairs bipartite but we also include the pairs for the elements that ought to be included in the beam

$$\ell(s, c) = \sum_{i=1}^{k} \sum_{j=i+1}^{n} \left( c_{\sigma^*(j)} - c_{\sigma^*(i)} \right) \max \left( 0, s_{\sigma^*(j)} - s_{\sigma^*(1)} + 1 \right) \tag{20}$$

The *weighted pairs (all)* loss provides many different variants as exemplified by *weighted pairs (bipartite)* and *weighted pairs (hybrid)*. We believe that exploring the ranking literature can lead to interesting insights on what losses to use for learning beam search policies in our framework.

# E  No-Regret Guarantees

This section presents analysis that leads to proofs of theorems 1, 2, 3, and 4. We analyze

$$c(\theta) = \mathbb{E}_{(x,y) \sim \mathcal{D}} \mathbb{E}_{\hat{y} \sim \pi(\cdot, \theta)} c_{x,y}(\hat{y}).$$

The prediction cost $c_{x,y}(\hat{y})$ is generated by sampling a beam trajectory $b_{1:h}$ with policy $\pi(\cdot, \theta)$. The prediction $\hat{y}$ is extracted from $b_h$. We have

$$c(\theta) = \mathbb{E}_{(x,y) \sim \mathcal{D}} \mathbb{E}_{b_{1:h} \sim \pi(\cdot, \theta)} \left( c^*(b_1) + \sum_{i=1}^{h-1} c(b_i, b_{i+1}) \right).$$

As $b_1$ depends only on $x \in \mathcal{X}$, $c^*(b_1)$ does not depend on the parameters $\theta$ and therefore can be ignored for optimization purposes. We analyze instead the surrogate

$$\ell(\theta, \theta') = \mathbb{E}_{(x,y) \sim \mathcal{D}} \mathbb{E}_{b_{1:h} \sim \pi(\cdot, \theta')} \left( \sum_{i=1}^{h-1} \ell(\theta, b_i) \right), \tag{21}$$

where $\ell(\cdot, b) : \Theta \to \mathbb{R}$ is a surrogate for $\mathbb{E}_{b' \sim \pi(b, \cdot)} c(b, b') : \Theta \to \mathbb{R}$. See Section 4.1 for extended discussion on the motivation behind surrogate loss $\ell(\cdot, b)$. It is convenient to assume that the policy $\pi(\cdot, \theta') : V_k \to \Delta(V_k)$ used to collect the beam trajectory $b_{1:h}$ can be different than the policy $\pi(\cdot, \theta) : V_k \to \Delta(V_k)$ used to evaluate the surrogate losses at the visited beams. The surrogate loss function $\ell : \Theta \times V_k \to \mathbb{R}$ depends on the sampled example $(x, y) \in \mathcal{X} \times \mathcal{Y}$, but we omit this dependency for conciseness.

## E.1  No-Regret Guarantees with Explicit Expectations

Here we present the proofs of Theorem 1 and Theorem 2. It is informative to consider the case where we have access to both explicit expectations. In this case, the no-regret algorithm is run on the sequence of losses $\ell(\theta_1, \theta_1), \ldots, \ell(\theta_m, \theta_m)$ yielding average regret

$$\gamma_m = \frac{1}{m} \sum_{t=1}^{m} \ell(\theta_t, \theta_t) - \min_{\theta \in \Theta} \frac{1}{m} \sum_{t=1}^{m} \ell(\theta, \theta_t).$$

As the sequence $\theta_1, \ldots, \theta_m$ is generated by a no-regret algorithm, the average regret goes to zero as $m$ goes to infinity. This result tells us that the uniform mixture obtained by sampling uniformly at random one of $\theta_1, \ldots, \theta_m$ and acting according to it for the full trajectory, is competitive with the best policy in $\Theta$ along the same induced trajectories. Note that

$$\frac{1}{T} \sum_{t=1}^{T} \ell(\theta_t, \theta_t) - \min_{\theta \in \Theta} \frac{1}{T} \sum_{t=1}^{T} \ell(\theta, \theta_t) = \mathbb{E}_{t \sim U(1,T)} \ell(\theta_t, \theta_t) - \min_{\theta \in \Theta} \mathbb{E}_{t \sim U(1,T)} \ell(\theta, \theta_t),$$

where $U(1, T)$ denotes the uniform distribution over $[T]$. Performance guarantees are obtained from the rearrangement

$$\frac{1}{m} \sum_{t=1}^{m} \ell(\theta_t, \theta_t) = \epsilon_m + \gamma_m,$$

where

$$\epsilon_m = \min_{\theta \in \Theta} \frac{1}{m} \sum_{t=1}^{m} \ell(\theta, \theta_t),$$

$$\gamma_m = \frac{1}{m} \sum_{t=1}^{m} \ell(\theta_t, \theta_t) - \min_{\theta \in \Theta} \frac{1}{m} \sum_{t=1}^{m} \ell(\theta, \theta_t).$$

Furthermore, if the surrogate loss $\ell(\cdot, b) : \Theta \to \mathbb{R}$ upper bounds the expected beam transition cost $\mathbb{E}_{b' \sim \pi(b, \cdot)} c(b, b') : \Theta \to \mathbb{R}$, i.e., $\ell(\theta, b) \geq \mathbb{E}_{b' \sim \pi(b, \theta)} c(b, b')$ for all $b \in V_k$ and all $\theta \in \Theta$, we have

$$\mathbb{E}_{b_{1:h} \sim \pi(\cdot, \theta)} \left( \sum_{i=1}^{h-1} c(b_i, b_{i+1}) \right) \leq \mathbb{E}_{b_{1:h} \sim \pi(\cdot, \theta)} \left( \sum_{i=1}^{h-1} \ell(\theta, b_i) \right),$$

and consequently,

$$\frac{1}{m} \sum_{t=1}^{m} c(\theta_t) \leq \frac{1}{m} \sum_{t=1}^{m} \ell(\theta_t, \theta_t) + \mathbb{E}_{(x,y) \sim \mathcal{D}} c^*(b_1),$$

i.e, we are able to use the expected surrogate loss incurred by the uniform mixture of $\theta_1, \ldots, \theta_m$ to upper bound the expected labeling cost resulting from labeling examples $(x, y) \sim \mathcal{D}$ with the uniform mixture of $\theta_1, \ldots, \theta_m$.

As the sequence $\theta_1, \ldots, \theta_m$ is chosen by a no-regret algorithm, $\gamma_m$ goes to zero as $m$ goes to infinity. The term $\epsilon_m$ is harder to characterize as $m$ goes to infinity. We are guaranteed that the uniform mixture of $\theta_1, \ldots, \theta_m$ and, as result the best policy in $\theta_1, \ldots, \theta_m$, is competitive with the best policy in hindsight $\theta_m^* \in \arg\min_{\theta \in \Theta} 1/m \sum_{t=1}^{m} \ell(\theta, \theta_t)$. For the performance guarantees to be interesting, it is necessary for $\epsilon_m$ to remain small as $m$ goes to infinity, i.e., there must exist a policy in $\Theta$ that performs well on the distribution of trajectories induced by the uniform mixture of $\theta_1, \ldots, \theta_m$. We think that this remark is often not adequately discussed in the literature. Nonetheless, for expressive policy classes, e.g., neural networks, it is reasonable to assume the existence of such a policy.

### E.2 Finite Sample Analysis

Next we provide a proof of Theorem 3. We typically do not have access to the explicit expectations in Equation (21). What we do have access to is an estimator

$$\hat{\ell}(\theta, \theta') = \sum_{i=1}^{h-1} \ell(\theta, b_i),$$

which is obtained by sampling an example $(x, y)$ from the data generating distribution $\mathcal{D}$, and executing policy $\pi(\cdot, \theta')$ to collect a trajectory $b_{1:h}$.

Our no-regret algorithm is then run on the sequence of sampled losses, yielding the sequence $\theta_1, \ldots, \theta_m$ and average regret

$$\hat{\gamma}_m = \frac{1}{m} \sum_{t=1}^{m} \hat{\ell}(\theta_t, \theta_t) - \min_{\theta \in \Theta} \frac{1}{m} \sum_{t=1}^{m} \hat{\ell}(\theta, \theta_t).$$

We show that the true population loss of the uniform mixture of $\theta_1, \ldots, \theta_m$ is, with high probability, not much larger than the empirical loss observed on the sampled trajectories, i.e.,

$$\mathbb{P}\left( \frac{1}{m} \sum_{t=1}^{m} \ell(\theta_t, \theta_t) \le \frac{1}{m} \sum_{t=1}^{m} \hat{\ell}(\theta_t, \theta_t) + \eta(\delta, m) \right) \ge 1 - \delta, \tag{22}$$

where $\delta \in (0, 1]$ is related to the probability of the statement, and $\eta(\delta, m)$ depends only on $\delta$ and $m$. Given this result, we are able to give performance guarantees for the uniform mixture of $\theta_1, \ldots, \theta_m$ as

$$\mathbb{P}\left( \frac{1}{m} \sum_{t=1}^{m} \ell(\theta_t, \theta_t) \le \hat{\epsilon}_m + \hat{\gamma}_m + \eta(\delta, m) \right) \ge 1 - \delta. \tag{23}$$

*Proof.* Define a function on beam trajectories. Assume that we have $0 \le \ell(\theta, b_{1:h}) \le u$, with $u \in \mathbb{R}$, for all $(x, y) \in \mathcal{X} \times \mathcal{Y}$ and for all beam trajectories $b_{1:h}$ through $G_k$, i.e., $b_1 = b_{(0)}$, $b_h \in T_k$, $b_i \in V_k$ for all $i \in [n]$, and $b_{i+1} \in N_{b_i}$ for $i \in [h-1]$. As a result, $0 \le \ell(\theta, \theta') \le u$ and $0 \le \hat{\ell}(\theta, \theta') \le u$, for all $\theta, \theta' \in \Theta$ and all $(x, y) \in \mathcal{X} \times \mathcal{Y}$. In our case,

$$\ell(\theta, b_{1:h}) = \sum_{i=1}^{h-1} \ell(\theta, b_i). \tag{24}$$

Construct the martingale sequence

$$z_t = \sum_{j=1}^{t} \left( \ell(\theta_j, \theta_j) - \hat{\ell}(\theta_j, \theta_j) \right), \tag{25}$$

for $t \in [m]$. It is simple to verify that the sequence $z_1, \ldots, z_m$ is a martingale, i.e., that we have $\mathbb{E}_{z_t | z_1, \ldots, z_{t-1}} z_t = z_{t-1}$ for all $t \in [m]$. Furthermore, we have $|z_t - z_{t-1}| \le u$ for all $t \in [m]$, where $z_0 = 0$. The high probability result is obtained by applying the Azuma-Hoeffding inequality to the martingale sequence $z_t$, for $t \in \mathbb{N}$, which yields

$$\mathbb{P}\left( \frac{1}{m} \sum_{t=1}^{m} \ell(\theta_t, \theta_t) \le \frac{1}{m} \sum_{t=1}^{m} \hat{\ell}(\theta_t, \theta_t) + u\sqrt{\frac{2\log(1/\delta)}{m}} \right) \ge 1 - \delta. \tag{26}$$

Revisiting Equation (23), for fixed $\delta \in (0, 1]$, as $m$ goes to infinity, we have that both $\hat{\gamma}_m$ and $\eta(\delta, m)$ go to zero, proving high probability no-regret guarantees for this setting. $\square$

### E.3 Finite Sample Analysis for Arbitrary Data Collection Policies

Finally, in this section, we provide a proof of Theorem 4. All the results stated so far are for the continue data collection strategy where, at time $t \in [m]$, the whole trajectory $b_{1:h}$ is collected using the current policy $\pi(\cdot, \theta_t)$. Stop and reset data collection strategies do not necessarily collect the

full trajectory under $\pi(\cdot, \theta_t)$. If a transition $(b, b') \sim \pi(\cdot, \theta_t)$ leads to a cost increase, then, the stop data collection strategy stops collecting the trajectory at $b'$, and the reset data collection strategy, the oracle policy $\pi^*(\cdot, c^*)$ is used to sample the transition at $b$ instead.

In this section, we relate the expected loss of $\pi(\cdot, \theta)$ on trajectories collected by a different policy $\pi'$ to the expected loss of $\pi(\cdot, \theta)$ on its own trajectories. Consider the following auxiliary lemma:

**Lemma 1.** *Let $f : X \to \mathbb{R}$ be a function such that $f(x) \in [a, a+r]$, for $a, r \in \mathbb{R}$ and $r \geq 0$ for all $x \in X$, that can be either discrete or continuous. Let $d, d'$ be two probability distributions over $X$. We have*

$$|\mathbb{E}_{x \sim d} f(x) - \mathbb{E}_{x \sim d'} f(x)| \leq r/2 ||d - d'||_1. \tag{27}$$

*Proof.* We prove the result for the case where $X$ is discrete, i.e., $d$ and $d'$ are discrete probability distributions. The result for discrete distributions is sufficient for our purposes. Let $|X| = e$, with $e \in \mathbb{N}$, then $d, d' \in \mathbb{R}^e$. We have

$$
\begin{aligned}
|\mathbb{E}_{x \sim d} f(x) - \mathbb{E}_{x \sim d'} f(x)| &= \left| \sum_{x \in X} d(x) f(x) - \sum_{x \in X} d'(x) f(x) \right| \\
&= \left| \sum_{x \in X} d(x)(f(x) - c) - \sum_{x \in X} d'(x)(f(x) - c) \right| \\
&= \left| (d - d')^T (f - c) \right| \\
&\leq ||f - c||_\infty ||d - d'||_1,
\end{aligned}
$$

where $c$ is an arbitrary constant in $\mathbb{R}$ and $f \in \mathbb{R}^e$ is the vector representation of the function. In the second equality, we use $\sum_{x \in X} d(x) = \sum_{x \in X} d'(x) = 1$. In the third equality, we express the expectations as inner products and slightly abuse notation by denoting the coordinate-wise subtraction of $c$ from $f$ as $f - c$. In the final inequality, we use the generalized Cauchy–Schwarz inequality for the pair of dual norms $|| \cdot ||_1$ and $|| \cdot ||_\infty$. The desired result is obtained by choosing $c = a + r/2$. $\quad\square$

Often, $\pi' = (1 - \beta)\pi(\cdot, \theta) + \beta \pi^*(\cdot, c^*)$ for $\beta \in [0, 1]$, i.e., a probabilistic interpolation of the learned policy and the oracle policy. We do a more general analysis that will be useful to provide regret guarantees for the stop and reset data collection strategies. It is not necessarily the case that, for a roll-in policy $\pi' : V_k \to \Delta(V_k)$, there exists $\theta' \in \Theta$ such that $\pi' = \pi(\cdot, \theta')$. We modify the notation in Equation (21) to capture this fact and write

$$\ell(\theta, \pi') = \mathbb{E}_{(x,y) \sim \mathcal{D}} \mathbb{E}_{b_{1:h} \sim \pi'} \left( \sum_{i=1}^{h-1} \ell(\theta, b_i) \right). \tag{28}$$

The roll-in policies $\pi' : V_k \to \Delta(V_k)$ that we consider induce distributions over beam trajectories in $G_k$ that have a component where the beam trajectory up to $h - 1$ can be thought as coming from $\pi(\cdot, \theta)$. For a policy $\pi'$ that is somehow derived from the learned policy $\pi(\cdot, \theta)$, we write $d_{\pi'} = \alpha(\theta, x, y) d_\theta + (1 - \alpha(\theta, x, y)) q$, where $d_{\pi'}$ is the distribution over trajectories induced by the roll-in policy $\pi'$, $d_\theta$ is the distribution over trajectories induced by the learned policy $\pi(\cdot, \theta)$, $q$ is the residual distribution over trajectories of the component that is not captured by $d_\theta$, and $\alpha(\theta, x, y)$ is the probability that the trajectory up to $b_{h-1}$ is drawn solely from $\pi(\cdot, \theta)$. For example, for the policy $\pi' = (1 - \beta)\pi(\cdot, \theta) + \beta \pi^*(\cdot, c^*)$, we have $\alpha(\theta, x, y) = (1 - \beta)^{h-2}$, where $\alpha(\theta, x, y)$ is independent of $\theta$ in this case. In this example, $\pi'$, at each step of the trajectory of length $h$, flips a biased coin and acts with probability $1 - \beta$ according to $\pi(\cdot, \theta)$ and with probability $\beta$ according to $\pi(\cdot, c^*)$.

**Relating expectations** We use Lemma 1 to relate $\mathbb{E}_{b_{1:h} \sim \pi(\cdot, \theta)} \left( \sum_{i=1}^{h-1} \ell(\theta, b_i) \right)$ and $\mathbb{E}_{b_{1:h} \sim \pi'} \left( \sum_{i=1}^{h-1} \ell(\theta, b_i) \right)$. We have

$$
\begin{aligned}
||d_{\pi'} - d_\theta||_1 &= ||\alpha(\theta, x, y) d_\theta + (1 - \alpha(\theta, x, y)) q - d_\theta||_1 \\
&= (1 - \alpha(\theta, x, y)) ||q - d_\theta||_1 \\
&\leq 2(1 - \alpha(\theta, x, y)),
\end{aligned}
$$

where we used that $||d_1 - d_2||_1 \leq 2$ for any two distributions $d_1, d_2$. Revisiting Equation (27), we have

$$\mathbb{E}_{b_{1:h} \sim \pi(\cdot, \theta)} \left( \sum_{i=1}^{h-1} \ell(\theta, b_i) \right) \leq \mathbb{E}_{b_{1:h} \sim \pi'} \left( \sum_{i=1}^{h-1} \ell(\theta, b_i) \right) + u(1 - \alpha(\theta, x, y)),$$

and as a result

$$
\begin{aligned}
\ell(\theta, \theta) &= \mathbb{E}_{(x,y) \sim \mathcal{D}} \mathbb{E}_{b_{1:h} \sim \pi(\cdot, \theta)} \left( \sum_{i=1}^{h-1} \ell(\theta, b_i) \right) \\
&\leq \mathbb{E}_{(x,y) \sim \mathcal{D}} \left( \mathbb{E}_{b_{1:h} \sim \pi'} \left( \sum_{i=1}^{h-1} \ell(\theta, b_i) \right) + u(1 - \alpha(\theta, x, y)) \right) \\
&= \ell(\theta, \pi') + u(1 - \alpha(\theta)),
\end{aligned}
\tag{29}
$$

where we defined $\alpha(\theta) = \mathbb{E}_{(x,y) \sim \mathcal{D}} \alpha(\theta, x, y)$, i.e., the probability of sampling the beam trajectory up to time $h - 1$ solely with $\pi(\cdot, \theta)$, or equivalently, the probability of $\pi(\cdot, \theta)$ incurring no cost increases up to time $h - 1$.

**Finite sample analysis with known schedules**  We now consider the finite sample analysis for the setting considered in this section. By arguments similar to those in Appendix E.2, we have

$$\mathbb{P} \left( \frac{1}{m} \sum_{t=1}^m \ell(\theta_t, \pi_t) \leq \frac{1}{m} \sum_{t=1}^m \hat{\ell}(\theta_t, \pi_t) + u \sqrt{\frac{2 \log(1/\delta)}{m}} \right) \geq 1 - \delta,$$

which, combining with Equation (29) implies

$$\mathbb{P} \left( \frac{1}{m} \sum_{t=1}^m \ell(\theta_t, \theta_t) \leq \frac{1}{m} \sum_{t=1}^m \hat{\ell}(\theta_t, \pi_t) + u \sqrt{\frac{2 \log(1/\delta)}{m}} + u \left( 1 - \frac{1}{m} \sum_{t=1}^m \alpha(\theta_t) \right) \right) \geq 1 - \delta,$$

$$\tag{30}$$

Equation (30) can be simplified for roll-in policies $\pi_t = (1 - \beta_t)\pi(\cdot, \theta) + \beta_t \pi^*(\cdot, c^*)$ with fixed interpolation schedules $\beta_t$, for $t \in \mathbb{N}$. For example, for $\beta_1 = 1$ for $t \in [t_0]$, for some $t_0 \in \mathbb{N}$, and $\beta_t = 0$ for $t > t_0$, we have

$$\mathbb{P} \left( \frac{1}{m} \sum_{t=1}^m \ell(\theta_t, \theta_t) \leq \frac{1}{m} \sum_{t=1}^m \hat{\ell}(\theta_t, \pi_t) + u \sqrt{\frac{2 \log(1/\delta)}{m}} + u \min \left( 1, \frac{t_0}{m} \right) \right) \geq 1 - \delta. \tag{31}$$

**Guarantees for the stop and reset data collection strategies**  The previous analysis allows us to provide regret guarantees for the reset data collection strategy. Steps in the trajectory are sampled using the learned policy $\pi(\cdot, \theta)$ when they do not result in cost increase, and sampled from $\pi^*(\cdot, c^*)$ otherwise, i.e., while sampling a trajectory $b_1, \ldots, b_i$ with $\pi(\cdot, \theta)$, if a cost increase would occur on the transition from $b_i$ to $b' \sim \pi(b_i, \theta)$, then rather than transitioning to $b_{i+1} = b'$, we transition to $b_{i+1} \sim \pi^*(b_i, c^*)$, and continue from $b_{i+1}$ until a terminal beam $b_h \in T_k$ is reached. In this case, $\alpha(\theta, x, y)$ is interpreted as the probability that the trajectory $b_{1:h-1}$ on the beam search $G_k$ induced by $x$ is sampled using only $\pi(\cdot, \theta)$, i.e., no cost increases occur up to time $h - 1$.

We can use this fact along with the previous results to obtain a regret statement for both the explicit expectation and the finite sample cases. The main difficulty is that $\alpha(\theta, x, y)$ and $\alpha(\theta)$ are not known. Again, the only way that we have access to them is through a sample estimate $\hat{\alpha}(\theta)$. We construct a martingale for this case involving both the randomness of the loss function and the reset probability.

We can use this information along with Azuma-Hoeffding inequality to give a joint concentration result. The martingale sequence that we now construct is

$$z_t = \sum_{j=1}^t \left( \ell(\theta_j, \pi_j) - \hat{\ell}(\theta_j, \pi_j) + u(1 - \alpha(\theta_j)) - u(1 - \hat{\alpha}(\theta_j)) \right), \tag{32}$$

which now includes the random variables of the estimator of the probability that we will reset at least once. Note that $\hat{\alpha}(\theta)$ also depends on $x, y, b_{1:h}$, which we omit for simplicity. Similarly to the

martingale arguments in Equation (25), Equation (32) defines a martingale. In this case, we have $|z_t - z_{t-1}| \leq 2u$ for all $t \in [m]$, and $z_0 = 0$. Applying Azuma-Hoeffding yields a result similar to Equation (30), i.e.,

$$\mathbb{P}\left(\frac{1}{m}\sum_{t=1}^{m}\ell(\theta_t, \theta_t) \leq \frac{1}{m}\sum_{t=1}^{m}\hat{\ell}(\theta_t, \pi_t) + 2u\sqrt{\frac{2\log(1/\delta)}{m}} + u\left(1 - \frac{1}{m}\sum_{t=1}^{m}\hat{\alpha}(\theta_t)\right)\right) \geq 1 - \delta,$$
(33)

Even if $1/m\sum_{t=1}^{m}\hat{\alpha}(\theta_t)$ remains at some nonzero quantity as $m$ goes to infinity, we can still give a guarantee with respect to this reset probability. Namely, if we observe that we are most of the time sampling the full trajectory with the learned policy, then we guarantee that we are not too far away from the true loss of the mixture policy.