[Reviews · NeurIPS 2018]

Reviewer 1



This is a carefully written paper framing learning beam search policies under imitation learning. The key insight to do this is to define a search space over beams rather than outpus, and considering actions that transition between beams. I liked how this idea allows us to conceptualize learning beam search policies and I think it will help other researchers in combining this with imitation learning, which is in my opinion a promising combination. The writing is dense, but very clear. I understand this is a theory paper. I would nevertheless I would have appreciated some experiments, even on synthetic data to demonstrate the potential benefits of taking beam search into account while learning incremental structured output predictors with imitation learning. Also, it would help illustrate what the various requirements in terms of oracle information and loss calculation might mean for a practical implementation. A few other points: - It is worth pointing out that the SEARN paper cited actually did use beam search in some of the imitation learning experiments, so the combination exists in the literature but it is under explored. - in equatlon 5, the rewriting of the loss with respect to the transitions does it affect the ability to handle non-decomposable loss functions? Do we assume that the loss increases monotonically? It would be good to clarify this as imitation learning algorithms such as SEARN are often able to learn with non-decomposable loss functions. - Section 5 mentions a 4th theorem that is not present in the main paper at least.

Reviewer 2



This paper is a purely theoretical paper on provable properties in learning with beam search via imitation learning. The basic setting is imitation learning, where you assume a "dynamic oracle" is available at any state. While this setting is definitely useful in some scenarios, e.g., syntactic parsing, it is not often available in harder problems such as machine translation. So while the theorems in this paper are interesting and went far beyond existing perceptron-style theorems, its real-world application is limited to those scenarios with easy-to-compute dynamic oracles. On the other hand, I think experiments in those imitation learning scenarios would be valuable to compliment the theoretical results.

Reviewer 3



Summary: This paper presents a general form of an algorithm for learning with beam search. This general form includes in it various learning to search methods in the literature -- early stopping perceptron, LaSo and Dagger. The paper also presents a regret analysis for this algorithm and shows no-regret guarantees for learning with beam search. Review: The generalization of various algorithms presented in the paper is interesting. Perhaps the primary contribution of the paper, however, is the no regret guarantee in section 5 -- previous guarantees for this setting are perceptron-style mistake bounds. This is a nice result. It is slightly surprising that the continue strategy (i.e. Dagger style imitation learning) has better guarantees than LaSo. I wonder if this makes a difference in practice though. However, the paper is notationally very dense and not easy to read. Somewhat frustratingly, a lot of the heavy lifting in the proofs have been moved to the appendix. For example, the paper talks about theorem 4, but even the statement of the theorem is in the appendix. It would have been good to see at least proof sketches in the body of the paper. Given that the primary contribution is a theoretical one, this part of the paper needs better explanation.

Reviewer 4



This paper develops a unifying formalism (a meta-algorithm) for imitation learning in the context of beam search; that is, learning in a beam-search-aware manner with access to an oracle that will tell you the minimum completion cost for any partial state. This formalism exposes a subtle but important corner of the space that has not yet been explored: algorithms that are beam aware, but which do not reset or stop after the gold-standard falls out of the beam, but instead continue along the best possible remaining path. Using this formalism, they are able to develop regret guarantees for this new case, alongside novel guarantees for existing cases where the algorithm stops or resets. They show the case where the algorithm continues to have better regret guarantees. This paper is a masterwork of notation. No symbol is wasted, no symbol is under-explained. Everything is crisp and precise. The paper is demanding, but with some patience, its clarity is remarkable. Its unifying framework builds an effective bridge between the worlds of beam-search aware structured prediction and imitation learning, and for this reviewer, that bridge is very valuable. The only thing I would have liked to see, which was missing from both the main paper and the appendices, is the inclusion of a table matching existing papers to specific loss-strategy combinations in the meta-algorithm. As implied in my summary above, the unexplored corner of imitation learning that they’ve uncovered with this formalism (that is, beam + continue with oracle costs), is novel and interesting. Even ignoring the regret guarantees, from an intuitive standpoint, it is likely to be useful for practitioners down the road, once we have some experiments supporting this choice (this paper has no experiments, which I assume is okay for NIPS). Some nit-picky concerns: The two points laid out in the first paragraph: (1) learning ignores the beam and (2) learning uses only oracle trajectories, are a good way to spell out the standard maximum likelihood training process. However, to say that your method is the only method the address both issues simultaneously is too strong. Any beam-aware method naturally addresses (1) and (2), as by definition it does not use *only* oracle trajectories. You need a better way of phrasing number (2) to emphasize that you are the proposing the first method to both use a beam and oracle costs to avoid stopping or resetting when the gold-standard falls out of the beam - which is a fair amount more subtle than being the first to use both a beam and non-oracle trajectories. This becomes clear two paragraphs later, but it is misleading for the first paragraph of the paper. The main text mentions Thm 4 without mentioning that it can only be found in the appendix. -- Dropping in to thank the authors for their response, and for including the table I requested.